# Aspiration Pneumonia with Prominent Alveolar Mineralization in a Dairy Cow

**DOI:** 10.3390/vetsci9030128

**Published:** 2022-03-10

**Authors:** Jasmine Hattab, Jessica Maria Abbate, Francesco Castelli, Giovanni Lanteri, Carmelo Iaria, Giuseppe Marruchella

**Affiliations:** 1Faculty of Veterinary Medicine, University of Teramo, Loc. Piano D’Accio, 64100 Teramo, Italy; jhattab@unite.it (J.H.); francesco.castelli@studenti.unite.it (F.C.); gmarruchella@unite.it (G.M.); 2Department of Veterinary Sciences, University of Messina, Polo Annunziata, 98168 Messina, Italy; jabbate@unime.it; 3Department of Chemical, Biological, Pharmaceutical and Environmental Sciences, University of Messina, Viale Ferdinando Stagno d’Alcontres 31, 98166 Messina, Italy; ciaria@unime.it

**Keywords:** dairy cow, tracheitis, aspiration pneumonia, alveolar calcification

## Abstract

A 2-years-old Jersey breed cow showed severe respiratory distress and prolonged lateral recumbency about 1 week after parturition. The cow was regularly vaccinated against the main respiratory pathogens and was given a calcium salt solution per os shortly after calving in order to prevent milk fever. Despite therapy with broad-spectrum antibiotics and anti-inflammatory drugs, the cow spontaneously died one week later and was necropsied. On gross examination, a severe, acute, diffuse fibrinonecrotic tracheitis was observed. In addition, the cranioventral portions of both lungs appeared firm and severely congested, while the pleural surface was covered by a discrete amount of fibrinous exudate. Microscopically, the following lesions were observed: tracheal hemorrhages, acute, fibrinonecrotic and suppurative tracheitis, pulmonary hemorrhages, fibrinous bronchopneumonia and fibrinous pleuritis. Noteworthy, multiple foci of mineralization were observed, scattered throughout the lung parenchyma and occasionally within the tracheal mucosa. The presence of calcium deposits was confirmed by means of Von Kossa staining method. Based on clinical history, clinical signs and pathological findings, aspiration pneumonia caused by the accidental inhalation of liquid calcium salt supplement was diagnosed. The present case report highlights the relevance of the staff training to optimize animal production and welfare.

## 1. Introduction

Respiratory diseases are one of the most common causes of economic losses in livestock, with a complicated and multivariate etiology resulting from complex interactions between pathogens (mainly viruses and bacteria), host (e.g., age and immune status) and environment (e.g., overcrowding, transport, inhalation of dust) [1,2]. Farm management plays a key role in determining the onset and severity of respiratory diseases, even though most diagnostic approaches mainly focus on microbiological investigations [1,3]. The term “bovine respiratory disease” (BRD) is commonly used in cattle to refer to a diverse heterogeneous group of severe disease conditions, which can affect animals of all ages and are caused by a variety of factors, either alone or in combination [1,4,5]. Such respiratory diseases may vary in pathophysiology, giving rise to different clinical and pathological findings, thereby making the veterinary practice as intriguing as it is challenging. Considering the above, gross and microscopic observations can be very useful for answering diagnostic queries, as well as for providing a mechanistic basis to better understand the etiology and pathogenesis of BRD [5]. Tracheitis is frequently caused primarily by viral agents (e.g., *bovine herpesvirus* 1, BoHV-1), which allow for secondary infections by opportunistic bacteria, affecting the clinical outcome [6]. Likewise, the inhalation of foreign material, such as ruminal content or iatrogenic depositions of chemical irritants, can also cause extensive inflammation and necrosis of the tracheal mucosa. Chemical-induced tracheitis can be severe enough to seriously affect the airflow, with significant consequences and respiratory dysfunction. Furthermore, foreign substance inhalation can damage the lung parenchyma, resulting in severe, often fatal aspiration pneumonia [4,5].

Herein, we describe the main clinical–pathological features of aspiration pneumonia, which was recently observed in a dairy cow after the inappropriate administration of liquid calcium salt supplement.

## 2. Case Presentation and Gross Findings

The event occurred on a dairy farm in Central Italy, which housed about 200 Jersey breed cattle. The average number of lactating cows was 105, with average daily milk production of around 15 L per cattle. Cows were fed with polyphite hay, corn, barley and soy, without any kind of silage. The herd achieved IBR-free status, while cows were regularly vaccinated against the main respiratory pathogens, using commercially available vaccines, namely bovine viral diarrhea virus (Bovilis^®^ BVD, MSD Animal Health; 5018 Upper Hutt, New Zealand), parainfluenza-3 virus, bovine respiratory syncytial virus and *Mannheimia haemolytica* A1 (Bovilis^®^ Bovipast RSP, MSD Animal Health; 5018 Upper Hutt, New Zealand).

Importantly, the farmer stated that parturient paresis (also known as “milk fever” or “hypocalcemia”) represented a major concern in this herd. Therefore, the oral administration of a liquid calcium salt supplement (Feedtech Ca-supplement, DeLaval; 3610, Pinetown, South Africa; total fluid volume = 2 L) was planned after parturition, aiming to prevent the onset of hypocalcemia during lactation.

In November 2021, a 2-year-old primiparous cow developed severe respiratory distress with tachypnoea, labored breathing, prolonged lateral recumbency and high fever (rectal temperature = 42 °C) about 1 week after calving. A similar case had occurred a couple of weeks earlier; in both cases, the administration of calcium supplement had been carried out improperly, as the cow containment had been particularly difficult.

Due to the non-specific clinical signs, the cow was treated with broad-spectrum antimicrobials (Izotricillina S C.M., IZO s.r.l., 25124 Brescia, Italy; 15 mL/die intramuscularly, corresponding to procaine benzylpenicillin 3,600,000 U.I. (International Unit) + dihydrostreptomycin sulphate 4.5 g) and steroidal anti-inflammatory drugs (Rapison, ATI s.r.l., 40064 Ozzano dell’Emilia, Italy; single administration of 10 mL intramuscularly, corresponding to dexamethasone 20 mg). Nevertheless, the clinical course rapidly worsened and the cow spontaneously died one week after the onset of clinical signs. Post-mortem examination was performed under field conditions by veterinary pathologists of the Faculty of Veterinary Medicine of the University of Teramo (Italy). The body condition score was good (4/5) and only mild post-mortem changes were evident. At necropsy, lesions only affected the lower respiratory tract. In more detail, the tracheal mucosa appeared diffusely hyperemic, edematous and covered with abundant fibrinonecrotic exudate, with multiple diphtheritic membranes exfoliating from the mucosal surface (Figure 1).

The cranioventral regions of both lungs were severely hyperemic and firm at palpation, with a discrete amount of fibrin adhering to the pleural sheets (Figure 2). Tracheobronchial and mediastinal lymph nodes were markedly enlarged, hyperemic and edematous in the cut section.

Representative samples of the trachea and lungs were collected, fixed in 10% neutral buffered formalin and routinely processed for histopathological examination. Additionally, bacteriological investigations were performed on lung lesions and yielded the isolation of a wide plethora of bacteria, which rapidly grew on different culture media (blood agar and MacConkey agar plates incubated for 48 h at 37 °C; Liofilchem; 64026 Roseto degli Abruzzi, Italy). No virologic investigation was carried out, since the herd achieved IBR-free status and animals were regularly vaccinated against the main viral respiratory pathogens. Additionally, based on clinical signs and pathological features, other causes of post-parturient death (i.e., endometritis) were ruled out. In particular, the post-partum involution of the uterus appeared normal, with no gross evidence of endometritis. The cow was young (primiparous) and both calving and delivery of the placenta occurred normally.

## 3. Histological and Histochemical Findings

Here, 5-μm-thick sections from formalin-fixed, paraffin-embedded tissues were stained with hematoxylin–eosin (H&E). Microscopically, the trachea appeared severely and diffusely inflamed and necrotic, with large amounts of fibrin and neutrophilic exudate embedding the entire thickness of the tracheal mucosa. The tracheal epithelium was thickened, hyperplastic, focally disrupted and showed squamous metaplasia. Large amounts of blood, fibrin and degenerating neutrophils were also observed, adhering to the tracheal surface. Scattered bacterial aggregates and plant material were seen throughout the necrotic debris (Figure 3).

Bronchi, bronchioles, alveoli and interlobular septa appeared diffusely and severely hyperemic, necrotic and hemorrhagic. Severe edema and multifocal fibrin deposits were observed within the airways and interlobular septa (Figure 4). Although detectable, bacterial aggregates and inflammatory cell infiltration were less evident in the alveolar spaces when compared to the tracheal mucosa.

Importantly, multiple foci of mineralization were observed, scattered throughout the lung parenchyma and occasionally within the tracheal mucosa. In particular, the alveolar spaces were mainly affected, appearing as calcified casts of this portion of the respiratory tract (Figure 5).

The presence of calcium deposits was confirmed by means of the Von Kossa staining method, which was carried out on 5-μm-thick tissue sections using nuclear fast red (Bio-Optica; Milano, Italy) for counterstaining (Figure 6).

Overall, the most relevant pathological findings were tracheal hemorrhages; acute, fibrinonecrotic and suppurative tracheitis; pulmonary hemorrhages; fibrinous bronchopneumonia and pleuritis; and alveolar calcification. Iatrogenic aspiration pneumonia caused by the incidental inhalation of calcium salt solution was diagnosed because the clinical history, clinical signs and pathological findings were considered. In this respect, the alveolar mineralization pattern was of particular importance to the diagnosis.

## 4. Discussion

Incidental inhalation of food or of other foreign substances is a rather common event that can severely impair the respiratory function in different ways, both in humans and animals, including through mechanical obstruction of the lower respiratory tract, inflammatory response induced by irritating chemicals and bacterial infections. In adult cattle, aspiration pneumonia due to the inhalation of the ruminal content can occur [4,5,6,7], usually resulting from swallowing disorders (“dysphagia”). As an example, tetanus-affected cows are more prone to aspiration pneumonia because of pharyngeal paralysis or paresis [8]. Similarly, the iatrogenic deposition of chemicals into the airways can induce mild-to-severe aspiration pneumonia, depending on the nature of such remedies [8]. This is likely to be an underestimated disease condition, as cattle breeders used to administer chemicals per os to prevent or treat metabolic disorders or infectious diseases. The administration of large volumes of fluids in too short a time by means of unsuitable equipment or drenching techniques can induce the development of iatrogenic aspiration pneumonia [9,10]. Importantly, hypocalcemia predisposes sufferers to the occurrence of iatrogenic aspiration pneumonia by impairing swallowing [5]; drenching cows in lateral recumbency can further increase the chance of accidental inhalation of remedies [9,10]. However, it should be noted that the cow under study appeared clinically healthy and that the present event resulted from the unsuitable drenching technique.

Clinical and pathological features of aspiration pneumonia are quite variable. The onset of clinical signs is often insidious (fever, productive cough, excessive mucus secretion), unless the inhaled material occludes the trachea causing instant death from asphyxia. Inflammation usually involves the tracheobronchial mucosa, the lung parenchyma and the pleural sheets, where fibrinous exudate accumulates [5,11]. In the present case report, the calcium salt solution severely damaged the tracheal mucosa, with pathological findings overlapping those described after accidental drenching of disinfectants [5]. Aspiration pneumonia commonly develops as an acute, necrotizing or gangrenous pneumonia, because of the irritative power of the inhaled materials and their ability to carry pathogenic or saprophytic bacteria [7,12]. The cranial lobe of the right lung is the most susceptible in cows due to anatomical reasons, as it is supplied by the accessory bronchus originating directly from the supracarinal trachea. However, the bilateral involvement of cranioventral lobes is not unusual [7,8], as in the case described herein.

Microscopic findings can provide useful information about the etiology of aspiration pneumonia, for example showing the presence of oil droplets or foreign body reactions [4,5,11]. The calcification of the lung parenchyma can be observed under several disease conditions in cows, showing different clinical and pathological features. Hypercalcemia due to the ingestion of calcinogenic plants—such as *Cestrum diurnum* and *Solanum malacoxylon*, which contain vitamin D analogs—induces a progressive wasting disease and the mineralization of soft tissues (called “metastatic calcification”), including pulmonary interalveolar septa. Moreover, dystrophic calcification can be observed within the necrotic core of granulomatous pneumonia by *Mycobacterium bovis*, as well as of degenerated hydatid cysts [4]. In the present case report, the microscopic pattern of mineralization was different when compared with the above scenarios. Calcium deposits along the alveolar epithelium and filling of the alveolar lumina strongly suggested a hypercalcemic microenvironment within the lower airways, further supporting the inhalation of calcium salts as the only plausible etiopathogenetic hypothesis.

## 5. Conclusions

In the authors’ opinion, the present case report provides some useful and practical “take-home messages”. First, diagnostic approaches exclusively focused on microbial pathogens may be useless, if not misleading, in cows affected by respiratory syndrome. The use of a detailed clinical history, along with pathological investigations, could be much more efficient to answer diagnostic queries and to solve farm-specific health issues. Moreover, this report highlights the importance of staff training to optimize animal production and welfare. Inappropriate daily practices can hamper farmers’ efforts to select and breed cows, causing avoidable economic losses. Finally, it should be noted that hypocalcemia can impair swallowing, meaning oral administration of remedies should be carefully considered and performed in cows whenever hypocalcemia is suspected. Therefore, alternative control strategies, such as the administration of feeding rations with a negative dietary cation–anion difference or low in calcium during the dry period, should be considered to effectively prevent milk fever [13].

## Figures and Tables

**Figure 1 vetsci-09-00128-f001:**
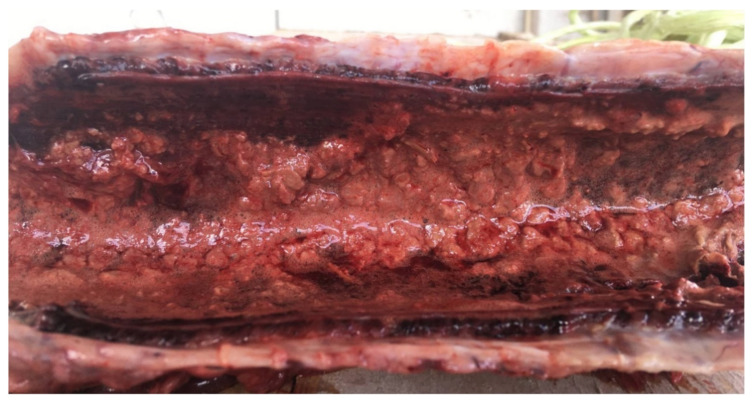
Tracheal mucosa. Severe, acute, diffuse fibrinonecrotic tracheitis. In the cut section, the tracheal wall appears strongly hyperemic, while the mucosa is diffusely covered by fibrinous–necrotic “bran-like” debris.

**Figure 2 vetsci-09-00128-f002:**
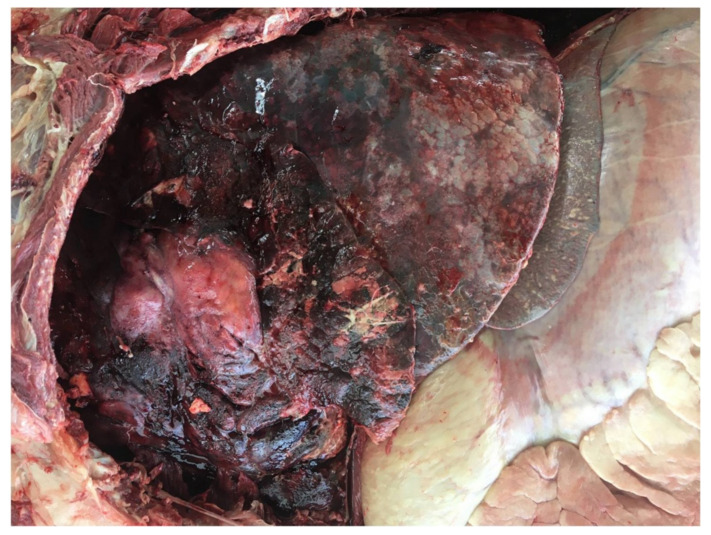
Left lung. Pleuritis and bronchopneumonia. Cranial and cardiac lobes appear congested and covered by discrete amounts of fibrin.

**Figure 3 vetsci-09-00128-f003:**
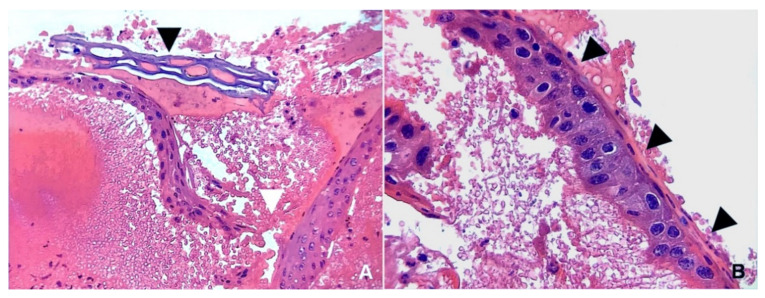
Trachea. (**a**) The lamina propria appears hemorrhagic and a large amount of blood also covers the tracheal epithelium. This appears focally disrupted (white arrowhead) and thickened. In detail, the tracheal epithelium is stratified, with surface cell layers being flattened (squamous metaplasia). Vegetable cells are evident and close to the tracheal mucosa (black arrowhead). Such findings support the incidental inhalation of foreign substances (HE, magnification ×100). (**b**) At a closer view, the squamous metaplasia of the tracheal epithelium (black arrowhead) can be better appreciated (HE, magnification ×200).

**Figure 4 vetsci-09-00128-f004:**
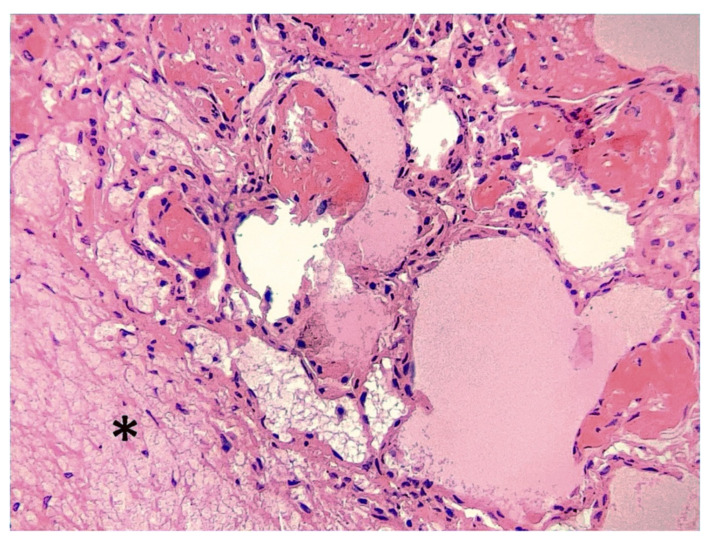
Lungs. Almost all alveoli are edematous or filled with fibrin. The interlobular septa (*) are thickened and embedded by abundant fibrinous exudate (HE, magnification ×200).

**Figure 5 vetsci-09-00128-f005:**
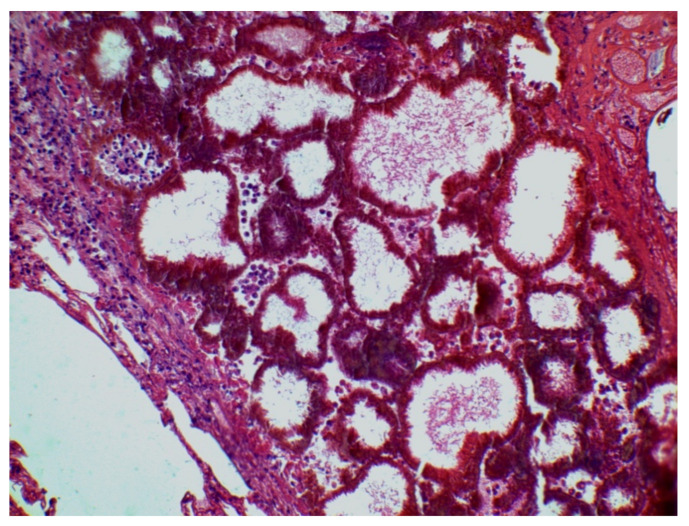
Lungs. The parenchyma appears widely hemorrhagic. Moreover, calcium deposits accumulate along the alveolar surface and partially obliterate the alveolar lumina (HE, magnification ×100).

**Figure 6 vetsci-09-00128-f006:**
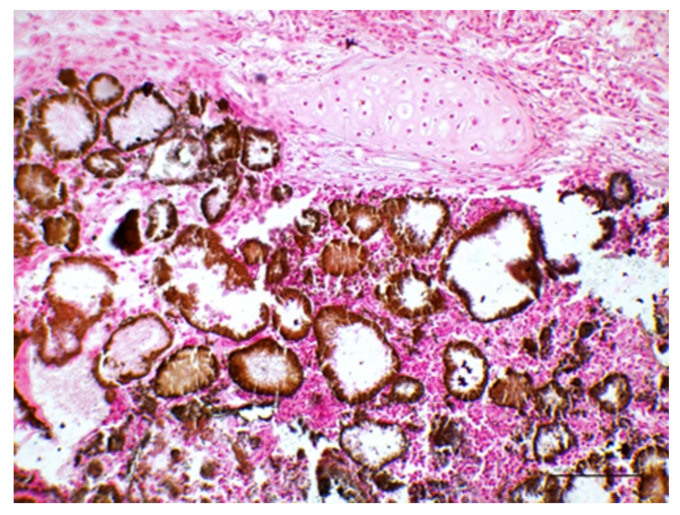
Lungs. Calcium salt deposits appear brown-black and are abundantly accumulated within the airways, particularly along the alveolar surfaces, thereby thickening the alveolar wall and partially obliterating the alveolar lumina (Von Kossa stain, magnification ×200, bar = 100 μm).

## Data Availability

Not applicable.

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
