# Peer review of "Aspiration Pneumonia with Prominent Alveolar Mineralization in a Dairy Cow"

_vetsci, 2022, doi:10.3390/vetsci9030128_

Round 1

Reviewer 1 Report

This type of bovine clinical cases are common in cattle herd, therefore his kind of clinical topics could help to Vets and students to learn practical cases to improve daily management practices to prevent animal diseases or death.

Also, there are some comments to check.

Author Response

Reviewer(s)' Comments to Author:

Reviewer: 1

This type of bovine clinical cases is common in cattle herd, therefore this kind of clinical topics could help to Vets and students to learn practical cases to improve daily management practices to prevent animal diseases or death.

Also, there are some comments to check.

Line 89. I cannot see black arrowheads….??? 

Authors’ response:

The Reviewer is right. In the end, we did not consider useful to indicate the tracheal wall with arrowheads. As a consequence, the sentence has been changed as follows: “…the tracheal wall appears strongly hyperemic, while the mucosa is diffusely covered by fibrinous-necrotic…”

Line 96. I think that Figure 2, it is not the best picture to represent diagnosis as pleuritis and bronchopneumonia. I consider that it is much better a general gross description of the thorax and their organs.

Authors’ response:

We do well understand and largely agree with the Reviewer’s concern. However, necropsy was carried out under field conditions, and this is the best picture we were able to take. We would like to maintain Figure 2 in the paper, as it adds some information about this kind of pneumonia. However, we are available to delete it, if necessary.

Lines 128,172. Find a synonym

Authors’ response:

The text has been changed according to the Reviewer’s suggestion.

Line 179. Asphyxia

Authors’ response:

The text has been changed according to the right Reviewer’s comment.

Lines 186-188. Why there is not a right lung picture, specially to shown cranial lobe?

 Authors’ response:

Unfortunately, we do not have picture showing the right lung. Due to the field conditions and to the necropsy technique (the cattle lay on its right side), we took “clean” pictures of the left side, before microbiological sampling.

Reviewer 2 Report

Dear Authors,

The case report of “Aspiration pneumonia with prominent alveolar mineralization in dairy cow” was review. The report described a case of cow that was improperly drenched with calcium supplement postpartum resulting in severe tracheitis and bronchopneumonia. The case history was well-described with proper macroscopic, microscopic findings and diagnostic conclusion. The main concern is regarding the novelty of this report as similar cases has been previously published. There is also little detailed discussion and regarding prevention methods of aspiration pneumonia when given calcium supplement to cattle with or without hypoglycemia. This is one of the main take home message for this case report and adding an update review on this subject matter is recommended.

Please see detailed comments/suggestions below;

  • Line 64-68, 71: Was the calcium supplement in liquid or paste form? What type of instrument was used in administration of the calcium supplement?
  • Please avoid mentioning of specific individual involvement in the clinical case.
  • In the discussion, it was noted that the cow did not have clinical signs of mild fever, but was not mentioned in the case history section.
  • Line 73: Also, please replace the term clumsily with improperly.
  • Postmortem examination: What was the body condition score of the animal and the degree of autolysis? Was uterus examined for evidence of endometritis? Were only the lung and trachea collected for histological examination? What tissues were collected for histological examinations? Was underlying IBR and BVD ruled out as the cause of tracheitis? It was mentioned in the history that the herd was IBR negative, but was it rechecked?
  • Line 101: Please list the bacteria isolated and approximate the amount of growth (low, moderate, high, heavy) if possible.
  • Line 110: Please use the term plant material instead of “vegetable cells”. Vegetable cells is used in human pathology referring to specific cell morphology in renal cell carcinoma.
  • Line 120-124: Please avoid using the term “lung parenchyma” for histological description and be more specific on which structure were affected (e.g. alveolar spaces, alveolar septa, bronchus, bronchioles, etc.).
  • What was the chronicity of the tissue lesions noted?
  • Fig 3 and 4 are not well white balanced.
  • May be add discussion on alternative administration methods or form of calcium supplement to avoid occurrences of aspiration pneumonia.

Author Response

Reviewer(s)' Comments to Author:

Reviewer: 2

Dear Authors,

The case report of “Aspiration pneumonia with prominent alveolar mineralization in a dairy cow” was review. The report described a case of cow that was improperly drenched with calcium supplement postpartum resulting in severe tracheitis and bronchopneumonia. The case history was well-described with proper macroscopic, microscopic findings and diagnostic conclusions. The main concern is regarding the novelty of this report as similar cases has been previously published. There is also little detailed discussion and regarding prevention methods of aspiration pneumonia when given calcium supplement to cattle with or without hypoglycemia. This is one of the main take home message for this case report and adding an update review on this subject matter is recommended.

Please see detailed comments/suggestions below;

Line 64-68, 71: Was the calcium supplement in liquid or paste form? What type of instrument was used in administration of the calcium supplement?

Please avoid mentioning of specific individual involvement in the clinical case.

In the discussion, it was noted that the cow did not have clinical signs of mild fever, but was not mentioned in the case history section.

Authors’ response:

The calcium supplement was liquid and (at the time of the present case report) it was administered by means of a plastic bottle. According to the right Reviewer’s comments, the sentences have been changed as follows: “Therefore, the oral administration of a liquid calcium salt supplement (Feedtech Ca-supplement, DeLaval, South Africa; total fluid volume = 2 liters) was planned after parturition, aiming to prevent the onset of hypocalcemia during lactation.”

“A similar case had occurred a couple of weeks earlier…”

Actually, as stated within the manuscript (lines 70-71), the cow developed high fever (…“high fever (rectal temperature = 42°C)…”).

Line 73: Also, please replace the term clumsily with improperly.

Authors’ response:

The manuscript has been changed according to the Reviewer’s comment.

Postmortem examination: What was the body condition score of the animal and the degree of autolysis? Was uterus examined for evidence of endometritis? Were only the lung and trachea collected for histological examination? What tissues were collected for histological examinations? Was underlying IBR and BVD ruled out as the cause of tracheitis? It was mentioned in the history that the herd was IBR negative, but was it rechecked?

Authors’ response:

The body condition score was “good” (4/5) and mild (early) postmortem changes were evident (necropsy was performed about 18 hours after death, during winter and with low environmental temperature). The post-partum involution of the uterus appeared normal, with no gross evidence of endometritis. The cow was “young” (primiparous) and both calving and delivery of the placenta normally occurred. As stated within the manuscript, we collected only lesions (trachea and lung) that we considered relevant for diagnostic purposes. The herd was and still is IBR-free, on the basis of regular serological tests. Considering history, clinical signs and pathological features, no further virological test was carried out. If necessary, such details could be added to the manuscript.

Line 101: Please list the bacteria isolated and approximate the amount of growth (low, moderate, high, heavy) if possible.

Authors’ response:

As stated within the manuscript, “a wide plethora of bacteria…rapidly grew…”. In such cases, we do not routinely proceed with the identification and titration of bacteria. If necessary, we could add this information in the text.

Line 110: Please use the term plant material instead of “vegetable cells”. Vegetable cells is used in human pathology referring to specific cell morphology in renal cell carcinoma.

Authors’ response:

The sentence has been changed as suggested by the Reviewer.

Line 120-124: Please avoid using the term “lung parenchyma” for histological description and be more specific on which structure were affected (e.g. alveolar spaces, alveolar septa, bronchus, bronchioles, etc.).

Authors’ response:

The manuscript has been changed as follows: “Bronchi, bronchioles, alveoli and interlobular septa appeared diffusely and severely hyperemic, necrotic and hemorrhagic. Severe edema and multifocal fibrin deposits were observed within the airways and interlobular septa (Figure 4). Although detectable, bacterial aggregates and inflammatory cell infiltration were less evident in the alveolar spaces when compared to the tracheal mucosa.”

What was the chronicity of the tissue lesions noted?

Authors’ response:

Overall, lesions appeared as acute-to-subacute.

Authors’ response:

Fig 3 and 4 are not well white balanced.

Authors’ response:

Pictures have been improved, as suggested by the Reviewer.

May be add discussion on alternative administration methods or form of calcium supplement to avoid occurrences of aspiration pneumonia.

Authors’ response:

The following sentence has been added at the end of conclusion: “Therefore, alternative control strategies, feeding rations with a negative dietary cation-anion difference or low in calcium during the dry period, should be considered to effectively prevent milk fever [13].” The cited reference has been added in the reference list.

Round 2

Reviewer 2 Report

Dear Authors,

Thank you responding and making the adjustments from the recommendations. Please see minor suggestions from your responses below; 

- Additional information of the gross findings (degree of autolysis, body condition of the animal), ruling out endometritis, and causes of tracheitis (IBR, mucosal disease) are pertinent information to reiterate in the content. It will indicate that throughout postmortem examination was performed to rule out common causes postparturient death and other causes of lesion that was observed. I highly suggest that it should be added to the content of this case report.

- Please state the chronicity, in this case acute to subacute, of the figure descriptions, and histological findings.

Author Response

Dear Editor,

Please find our Response to Referee, point-by-point, regarding the manuscript ID vetsci-1624392 entitled "Aspiration pneumonia with prominent alveolar mineralization in a dairy cow" by Hattab J et al.

We have addressed all concerns, detailed answers to reviewers’ comments are provided below and all corrections have been tracked in the manuscript.

We hope that the revised manuscript is now suitable for publication in Veterinary Sciences journal.

On the behalf of all Authors,

Yours sincerely,

Giovanni Lanteri

Reviewer(s)' Comments to Author:

Reviewer: 2

Dear Authors,

Thank you responding and making the adjustments from the recommendations.

Please see minor suggestions from your responses below; 

Additional information of the gross findings (degree of autolysis, body condition of the animal), ruling out endometritis, and causes of tracheitis (IBR, mucosal disease) are pertinent information to reiterate in the content. It will indicate that throughout postmortem examination was performed to rule out common causes post parturient death and other causes of lesion that was observed.

I highly suggest that it should be added to the content of this case report.

Authors’ response.

Additional information has been added as suggested by the Reviewer.

Please state the chronicity, in this case acute to subacute, of the figure descriptions, and histological findings.

Authors’ response.

The sentences have been changed accordingly throughout the text, as suggested.
